# Application of the Electromagnetic Method to the Spatial Distribution of Subsurface Saline and Fresh Water in the Coastal Mudflat Area of Jiangsu Province

**DOI:** 10.3390/s23146405

**Published:** 2023-07-14

**Authors:** Wei Zhu, Wenguo Wang, Dayong Wang, Gang Wang, Aiming Cui, Yongzai Xi, Fei Li, Baowei Zhang, Gege Zhang

**Affiliations:** 1Institute of Geophysical and Geochemical Exploration, Chinese Academy of Geological Sciences, Langfang 065000, China; wwenguo@mail.cgs.gov.cn (W.W.); cgswgang@mail.cgs.gov.cn (G.W.); caiming@mail.cgs.gov.cn (A.C.); xyongzai@mail.cgs.gov.cn (Y.X.); lfei@mail.cgs.gov.cn (F.L.); zhanggege@mail.cgs.gov.cn (G.Z.); 2National Center for Geological Exploration Technology, Langfang 065000, China; 3Key Laboratory of Geophysical Electromagnetic Probing Technologies, Ministry of Natural Resources, Langfang 065000, China; 4Terrestrial Geophysical Research Center, China Geological Survey, Langfang 065000, China; 5Geophysical Survey Center, China Geological Survey, Langfang 065000, China; zbaowei@mail.cgs.gov.cn

**Keywords:** coastal Jiangsu, mudflat area, subsurface structure, electromagnetic exploration, saline and fresh water distribution

## Abstract

Interfacial zones straddling terrestrial and marine realms, colloquially known as mudflats, epitomize a dynamic nexus between these environments and are fundamental to the coastal ecosystem. The investigation of these regions is paramount for facilitating infrastructural developments including ports, wharfs, cross-sea bridges, and the strategic utilization of freshwater resources sequestered from mainland islands amid ongoing economic progress. Terrestrial realms conventionally employ electromagnetic techniques as efficacious modalities to delineate subterranean geological information, encompassing structural details and water-bearing strata. However, the peculiar topographic and geological nuances of mudflat regions pose substantial challenges for the efficacious application of electromagnetic methodologies. The present paper endeavors to address these challenges by suggesting innovative modifications to the existing instrumentation and evolving novel data acquisition techniques specifically tailored for electromagnetic exploration within mudflat environments. This paper delves into the electrical characteristics of water-bearing layers within mudflats, and ascertains details pertaining to the subterranean structure and the spatial distribution of fresh and saline water resources, through the holistic interpretation of a multitude of profiles.

## 1. Introduction

Techniques regularly employed for investigating the intertidal zone (the region inundated by tides) encompass multibeam bathymetry, single-beam echosounding, side-scan sonar, shallow seismic profiling, and single-channel seismic surveying. These methodologies contribute to the study of the seabed’s topographical and geological properties, the dispersal of sedimentary rocks and layers, superficial constructions, fault lines, shallow gas, sand undulations, as well as potential geological hazards encompassing landslides, subsidence, and uplift. These techniques further elucidate the distribution of paleo-river channels and sedimentary environments. In addition to these techniques, remote sensing methodologies are implemented to observe the spatiotemporal alterations of the intertidal coastlines, the process of mudflat erosion and sedimentation, alterations in subaqueous topography, proximal suspended sediment data, and other mudflat-associated geological and environmental information [1,2,3,4,5,6,7].

Nonetheless, geophysical investigations into the intertidal terrain of mudflats are scarcely encountered. Within the proximal intertidal region of the Mekong River estuary, Thailand has conducted rudimentary seismic explorations, primarily for the detection of geological risks, including submarine landslides and environmental contamination. For instance, in 2013, Befus and Bayani undertook DC resistivity surveys within the mudflat region of New Zealand’s Cook Islands, obtaining insights into the aquiferous characteristics of strata extending beyond 20 m in depth. In 2003, Best utilized time-domain electromagnetic (TDEM) and ground-penetrating radar (GPR) techniques to assess a delta in Canada, thereby approximating the depths of the upper, intermediate, and lower sedimentary layers, along with the bedrock and the freshwater layer’s depth within the intertidal zone. At the southern flank of Hangzhou Bay in Zhejiang, China, various exploration methodologies were evaluated in the context of constructing a transmarine bridge, encompassing gravity measurements, shallow seismic profiling, and shallow geological section assessments. An exhaustive analysis of these methodologies provided insights into the sedimentary features, superficial structures, and the spread of fault lines in the shallow layers. The rudimentary seismic technique was also applied to ascertain the distribution of shallow gas, a potential peril to engineering projects [8,9,10]. Such studies are predominantly concentrated on the coastal terrestrial zone or shallow aquatic region, and the restricted depth of detection coupled with the confined methodology results in a narrow scope of underground geological data acquired.

The electric properties of subterranean strata in mudflat regions, encompassing both non-water bearing and water-bearing strata with varying mineralization degrees, present considerable variations, thus providing a fundamental prerequisite for conducting electromagnetic investigations in these zones. The merits of electromagnetic methodologies in the coastal domain, such as their capacity for deeper exploration, high discernibility of low-resistivity layers, and economic efficiency, make them suitable for detecting more profound geological data in regions with extensive sedimentary coverage. Nevertheless, owing to the unique topographical attributes of mudflat areas, electromagnetic explorations in the coastal regions have predominantly been restricted to employing land-based electromagnetic equipment and collection methods in reclaimed zones adjacent to mudflats, with a principal focus on sea intrusions, prehistoric river channels, and geothermal resources. By assessing the existing work and its outcomes, it becomes evident that there are negligible differences in the equipment, implementation, and interpretation techniques between terrestrial electromagnetic surveys conducted in reclaimed regions and traditional land-based electromagnetic surveys. Consequently, these methodologies are deemed inappropriate for utilization in mudflat regions [11,12,13]. In the initial phases, mudflat zones were primarily employed for marine reclamation and aquaculture, with no significant requirement for extensive engineering construction, which, in turn, has constrained the advancement of geotechnical investigation methodologies and techniques pertinent to shallow mudflats. As illustrated in Figure 1, conventional terrestrial electromagnetic exploration methodologies are implemented in terrestrial regions, and marine electromagnetic exploration techniques are utilized in seawater areas with depths exceeding 3 m. However, in the intertidal zone, these traditional terrestrial and marine exploration methodologies prove ineffective in conducting efficacious operations. Aspects such as amphibious work platforms, transmitter layouts, waterproof and corrosion-resistant equipment, efficient data collection methods, and interpretation techniques specifically tailored for electromagnetic exploration in mudflat regions remain largely uncharted territory.

This study endeavors to tackle the challenges inherent in electromagnetic exploration within mudflat regions. To begin with, we refined conventional electromagnetic equipment to surmount the restrictions imposed by existing methodologies. Subsequently, we carried out experimental procedures on a highly efficient data collection technique, reliant on a water–land amphibious platform employing a towed system. This system facilitated the acquisition of various electromagnetic datasets apt for mudflat regions, such as controlled-source audio-frequency magnetotellurics (CSAMT), audio-frequency magnetotellurics (AMT), and magnetotellurics (MT). Utilizing two-dimensional inversion processing, we extracted the subterranean electrical properties of the mudflat area based on multiple electromagnetic profiles acquired east of Xiaoyangkou (Figure 2a), Nantong, in Jiangsu Province. We amalgamated the inversion outcomes with pertinent data, encompassing drilling, hydrogeological, and shallow seismic data, to deduce the distribution of underground structures and fresh and saline water within the study region. The resultant findings are expected to supply foundational geological data to facilitate major engineering construction and freshwater resource development in the area. The positions of the electromagnetic profile layout, the CSAMT transmission source, and the collection points for the drilling data are depicted in Figure 2b.

## 2. Study Area Overview

### 2.1. Landform Overview

Positioned on the northern flank of the Yangtze River Economic Belt, Jiangsu Province comprises merely 1% of China’s total land area. However, it is teeming with mudflat and marine resources, spreading across roughly 5000 km^2^, an expanse that constitutes a quarter of China’s total mudflat area, thereby showcasing significant potential for development and utilization. The intertidal zone in Nantong is characterized by a vast mudflat region shallower than 0 m extending up to 15 km seawards, with a distribution that aligns nearly parallel to the coastline. In the deeper waters, well-established tidal sand ridges and channels are present, converging in a fan-like formation centered around Xiaoyangkou. These sand ridges are interspersed with deep channels featuring steep slopes and diverse water depths. The slope of the mudflat varies from 0.5 to 1.2%, with the mudflat surface being primarily flat, interspersed with some shallow and linear tidal channels. Certain mudflat areas, such as Lengjiasha and Dongtaiyangsha, are situated distant from the coastline. The sediments predominantly comprise grey and grey-yellow fine sand intercalated with mud. The nearshore shallow water areas, influenced by the complex coastal and estuarine topography, typically exhibit a larger tidal range at the bay head than at the bay mouth. With a considerable tidal range, the coastal areas experience an average tidal variation of 1.5 to 3.7 m, with the maximum tidal range reaching up to 3.95 m.

### 2.2. Geological Overview

The area of interest is situated in the northeastern quadrant of the Yangtze block and the northern boundary of the Nantong uplift, bordering the Dongtai sag to the north and east, falling within the jurisdiction of the Yangtze block. The region is extensively draped by unconsolidated sediments of the Quaternary period, with the bedrock being the Pukou Formation’s sandstone (K_2_p) from the Cretaceous period, extending to depths of approximately 800–1400 m and primarily composed of Neogene sediments in a sedimentary basin. The Quaternary system is well-established, spanning from the lower Pleistocene to the Holocene epochs. The lower stratum primarily consists of clay and sub-clay from grey-green, brown-red, and grey-yellow lacustrine deposits, interlayered with fine sand, and fine to coarse fluvial sand deposits in the sag. The upper stratum chiefly comprises medium to coarse grey-white and grey-green fluvial sand deposits. The Quaternary sediments within the study area boast abundant sources and robust deposition, particularly in the Nantong region proximal to the mouth of the Yangtze River, where the thickness of the Quaternary sediments can reach 200–360 m [14]. Influenced by the evolution of ancient rivers, periodic climatic oscillations from cold to warm, as well as marine transgressions and regressions, the Quaternary deposits primarily constitute an enormously thick loose layer with multiple sedimentary rhythms from alternating marine and terrestrial phases [15]. This is characterized by a stratified structure consisting of continuous clay and sand layers, leading to the formation of multiple confined aquifers within the tidal flat areas. This geological configuration facilitates the processes of freshwater dilution and seawater refilling [16].

Within the research area, the phreatic layer, the first confined aquifer Ⅰ, and the second confined aquifer Ⅱ are interconnected due to the thin or absent separating layers between them. Moreover, influenced by different episodes of marine transgressions, the groundwater has been salinified. Despite later dilution through infiltration from the Yangtze River and atmospheric precipitation, it still contains a high proportion of seawater salts, with a mineralization degree above 10 g/L. The water chemistry is typically of a Cl–Na type, and the resistivity values generally exhibit low-resistivity characteristics.

The third confined aquifer Ⅲ has favorable burial conditions. The top stratum is composed of dense brown-yellow sub-clay, which is stable in distribution and has a large thickness, effectively blocking the saline water from the overlying first and second confined aquifers. Therefore, within this area, the water quality of the third confined aquifer is significantly different from the upper confined aquifers. Most regions have a mineralization degree less than 1 g/L, indicating that the water is freshwater. The water chemistry is primarily of an HCO3-Ca·Na type, and compared to saline water-containing strata, the resistivity values exhibit high-resistivity characteristics [17].

### 2.3. Geoelectric Characteristics

The groundwater within the study area predominantly resides in the Neogene and Quaternary unconsolidated sedimentary sand layers, boasting an overall thickness exceeding 500 m and maximum depths within the 750–1000 m range. The electrical attributes of the formations are primarily dictated by the development level of the water-bearing sand layer or ancient river channel, and the salinity of the groundwater. Freshwater sand layers present relatively high resistivity, whereas those of saline water sand layers display low resistivity. The thickness of the saline water layer within the Quaternary formations surpasses 300 m, showcasing an exceptionally low resistivity of just a few ohm-meters. Based on the evaluation of existing drilling data from the vicinity of the study area, the Quaternary formations and their electrical characteristics align with the regional geology. The aquifers are primarily constituted of fine sand or silt layers exhibiting high resistivity, and the resistivity difference between the aquifers and the encompassing clay layers is approximately 10–20 Ω·m, with the depth of the aquifers exceeding that reflected in the regional data [18].

Notable disparities exist in resistivity among different formations and rock types within the area, fluctuating in a pattern akin to that of density. Typically, the resistivity of lithified formations surpasses that of unconsolidated layers. Within a certain depth, there are two types of geoelectrical structure patterns within the region: freshwater and saline water zones. Based on the differences in resistivity (ρ_s_), under the freshwater geoelectrical structure pattern, there is a five-layer electrical stratification feature with ρ1 < ρ2 > ρ3 > ρ4 << ρ5. In the saline water geoelectrical structure pattern, there is a four-layer electrical stratification feature with ρ1 > ρ2 < ρ3 << ρ4. The range of resistivity (ρ_s_) values and geological attributes corresponding to the resistivity layers ρ1, ρ2, ρ3, and ρ4 under different patterns are shown in Table 1 [19].

Furthermore, as tidal levels in the coastal zone shift, the modification of vertical conductivity in groundwater proceeds as follows: the conductivity remains relatively static from the groundwater surface to a predetermined depth. Beyond this point, the conductivity escalates rapidly, signifying the upper boundary of the freshwater–seawater mixing zone. These observations suggest that the physical conditions necessary for the electromagnetic method to ascertain the distribution of saline and fresh groundwater in the tidal zone are indeed present [20,21].

## 3. Data Collection

The AMT data collection was executed employing the V5 networked multifunctional electric instrument (hereafter referred to as V5), a product of the Canadian Phoenix Geophysics Company. Meanwhile, the CSAMT made use of the V8 multifunctional electric instrument, also manufactured by the same enterprise. In light of the high salinity environment and the condensed work window imposed by tidal conditions within the intertidal zone, precedents from previous intertidal zone investigations were utilized, and the electric and magnetic sensors were accordingly modified for data capture. Following these adjustments, a towed observation system was constructed and hinged on an air-cushioned vessel platform.

### 3.1. Instrument Modification

Non-polarized electrode receiving array: The conventional non-polarized electrode employs vertical bottom contact, which can easily topple and suffer from seawater corrosion in the beach region, thereby affecting the stability of electric signal reception. To address this, we have designed a horizontally positioned multipoint attachment non-polarizing electrode as depicted in Figure 3. The towing cable is linked to the electrode within the reserved pipe in the electrode body’s middle, piercing out at both ends. The reserved pipe is then injection-molded and sealed to ensure waterproofing and corrosion resistance. Multiple integrated non-polarized electrodes are linked to the towing cable at predetermined electrode distances to form the non-polarized electrode receiving array. During the towing process, the cable and non-polarized electrode on the receiving array are influenced by gravity. This results in the towing process being tightly intertwined with the mud and sand on the beach surface, thereby enhancing the stability of the received electric signal.

Magnetic sensor: Traditional land-based magnetic sensors are structured with non-waterproof tops and bottoms, and their signal output connectors and cables lack a waterproof design. When utilized in mudflat areas, seawater intrusion into the internal sensor can trigger its functional failure, thereby impeding data collection. The waterproof magnetic sensor, as illustrated in Figure 4, encompasses a magnetic sensor body and a magnetic sensor shell that creates an encapsulating cavity. The magnetic sensor body is installed within this cavity, and waterproof covers are fitted at the top and bottom of the magnetic sensor shell. A cable waterproof cap is installed at one end of the magnetic sensor shell, and the waterproof cable connector is equipped with a cable waterproof screw. The magnetic sensor body comprises interconnected coils, cores, and circuits, which are filled with insulating potting glue to amplify the waterproof properties of the magnetic sensor body. This design ensures that even if the upper and lower waterproof covers degrade due to the aging of rubber gaskets and O-rings, the internal magnetic sensor body can maintain normal operations. Thus, corrosion of internal electronic components that could affect the stability of magnetic signal reception is prevented.

### 3.2. Drag-and-Drop Mobile Reception System

Drawing upon the observational apparatus employed in marine CSEM, we adopted an electromagnetic separation approach based on conventional magnetotelluric measurement equipment. This involves positioning the magnetic path fixed point near the survey line or setting up along the vertical direction of the survey line on the shore for observation (Figure 5). The electric path employs a non-polarized electrode receiving chain for observations made parallel to the survey line direction. One measuring array utilizes a receiving chain composed of a multicore electrical cable connected to four non-polarized electrodes, with 25 m between two electrodes. Upon completing one observational array, an air-cushioned ship’s movable platform drags the non-polarized electrode receiving chain, moving it along the survey line direction to a specified station location to commence fixed-point observations. The system employs scalar observation, with the electric field *Ex* observed in line direction and the magnetic field *Hy* observed in the line-perpendicular direction.

The incorporation of integrated electric and magnetic sensors along with non-polarized electrode receiver chains mitigates the issue of seawater corrosion, thereby enhancing the stability of both electric and magnetic signal reception. Simultaneously, the employment of a mobile towing receiver system addresses the problem of limited survey efficiency resulting from personnel constraints and swift tidal changes in mudflat regions. As shown in Figure 2b, the transmitting and receiving distance (r) of CSAMT (survey lines L3, L4, L7) ranges from 5.5 to 7 km, with a transmitter dipole distance (AB) of 1 to 1.5 km. Figure 6a presents a record of unstable high-frequency transmission currents in the CSAMT under a low-resistance observation environment in the tidal flat area. Figure 6b depicts a record of stable high-frequency transmission currents after implementing deep excavation of the transmission electrode pit and backfilling with dry soil. The observed frequency range for CSAMT is 0.125 to 8192 Hz (observation duration not less than 1 h), while for AMT, it is 10,000 to 0.35 Hz (observation duration not less than 1 h). Figure 7 presents the electromagnetic data gathered from the mudflat area. Notably, the signal-to-noise ratio of the AMT measurement data dips beyond frequencies less than 5 Hz, and the data quality of certain AMT measurement points suffers due to the effects of the dead band within the frequency range of 1000 Hz–5000 Hz. Nevertheless, stable and high-quality observational data can be obtained in other frequency bands. Therefore, to compensate for the deficiencies in the data quality of the AMT high-frequency band, it becomes necessary to undertake CSAMT measurements at those AMT measurement points where high-frequency data are poor [22].

## 4. Methodology

In this study, the electric field Ex and magnetic field Hy data of CSAMT and AMT quadrature collected using a towed mobile receiving system in Rudong, Nantong, Jiangsu, were calculated using Equation (1) to obtain the Cagniard resistivity [23,24].
(1)$$\rho =\frac{1}{\omega \mu }{\left|\frac{E_{x}}{H_{y}}\right|}^{2},$$
where μ is the magnetic permeability, and H/m. ω is the circular frequency of the harmonic current, rad/s. $E_{x}$ is the field value of the x-component of the electric field on the surface of a uniform half-space in the right-angle coordinate system, V/m. $H_{y}$ is the field value of the y-component of the magnetic field on the surface of a uniform half-space in the right-angle coordinate system, A/m.

Using the OCCAM inversion method, seven inversion profiles were obtained. To suppress model structure not required by the data, the model roughness must be minimized. For a 2D structure with x in the direction of the strike axis, a measure of the model roughness [25] may be given by:(2)$$R_{1}={\left\|\partial_{y}m\right\|}^{2}+{\left\|\partial_{z}m\right\|}^{2},$$
where m is the vector of model parameters, $\partial_{y}m$ is a roughening matrix which differences the model parameters of laterally adjacent prisms, and $\partial_{z}m$ is a roughening matrix which differences the model parameters of vertically adjacent prisms.

In general inversion methods, the Earth model is first assumed to be $m=[\;m_{1},\;m_{2},\;\ldots ,m_{M}]$, the N observation data are $d=[\;d_{1},\;d_{2},\;\ldots ,d_{N}]$, and the estimated deviation is $e=[\;e_{1},\;e_{2},\;\ldots ,e_{N}]$. The fit of the theoretical model response $F\left[\;m\right]$ to the observation data can be expressed as:(3)$$X_{d}^{2}=\left(\;d-F\left[\;m\right]\right){\;}^{T}C_{d}^{-1}\;\left(\;d-F\left[\;m\right]\right),$$
where $C_{d}$ is known as the data covariance matrix. Nowadays, most MT inversion schemes solve non-uniqueness by finding the minimal possible model structure for a given fit level, thus stabilizing the inversion. Consider the general form of the model norm:(4)$$X_{m}^{2}=(\;m-m_{0}){\;}^{T}C_{m}^{-1}\;(\;m-m_{0}),$$
where $m_{0}$ is the initial model, and $C_{m}$ is the covariance matrix of the model. The minimum structure inversion problem is to minimize $X_{d}^{2}=X_{*}^{2}$ under condition $X_{m}^{2}$, where $X_{*}^{2}$ is the expected fit level.

The objective and penalty functions of the OCCAM method were obtained after using the roughness matrix instead of the model parametrization in the general inversion method.
(5)$$U\left(\;m,\lambda \right)=R_{1}+\lambda^{-1}\left\{\;\left(\;d-F\left[\;m\right]\right){\;}^{T}C_{d\;}^{-1}\left(\;d-F\left[\;m\right]\right)-X_{*}^{2}\;\right\},$$
(6)$$W_{\lambda }(m)=R_{1}+\lambda^{-1}\{\;(d-F[\;m]){\;}^{T}C_{d\;}^{-1}\;(d-F[\;m])\;\}\;$$
where $R_{1}$ is the model roughness, d is the observation data, $F\left[\;m\right]$ is the theoretical model response, and $\lambda^{-1}$ is the Lagrange multiplier.

The characteristic of OCCAM is that in each iteration the parameter λ is used not only as a step control parameter but also as a smooth parameter. In each iteration, a univariate search for λ is performed to find the model with the minimum data fit difference as the model for the next iteration until the desired level of fit difference is reached.

In light of the geological and borehole data available in the study area, we identified a strong correlation between the resistivity transition interface, from high to low, observed on the profile and the water content of saline and fresh water in the subsurface strata. By depth-correcting the apparent resistivity, we procured five categories of corrected resistivity values and, with distinct depth constraints, determined the corresponding aquifer thicknesses for these respective resistivity ranges. Furthermore, by integrating the detailed stratigraphic information extracted from the shallow seismic measurement profile, we employed the hydrogeological stratification information from the collective interpretation of electromagnetic, shallow seismic, and offshore single-channel seismic profiles distributed on the plane. This was instrumental in establishing the bottom boundaries of various features such as the brackish water-dominated unconfined aquifer layer, brackish water-dominated Ⅰ pressurized aquifer, brackish water-dominated Ⅱ pressurized aquifer, fresh water-dominated Quaternary, and the Ⅲ pressurized aquifer, by utilizing the Kriging interpolation method. This allowed for the accurate delineation of the distribution traits of different aquifers in the longitudinal direction, furnishing quantitative results with exceptional spatial resolution for studies of seawater intrusion and unconfined aquifer desalination. This valuable data will be instrumental in guiding the exploitation and utilization of water resources within the study area.

## 5. Profile Interpretation Method

### 5.1. Drill Hole Data Analysis

Utilizing borehole data acquired courtesy of the Jiangsu Yangtze River Geological Survey Institute, we constructed a stratigraphic cross-section (as depicted in Figure 8), correlating to two segments indicated by the black lines within Figure 2b. This borehole cross-section reveals the existence of a roughly 130 m thick layer of coarse sand and gravel in the Quaternary system’s lower section, while the Neogene system’s upper portion is characterized by a clay sand aquitard of approximately 120 m thickness. This suggests that the study area exhibited a consistent sedimentary environment from the late Neogene into the early Quaternary periods, with primary sediment components being fine-grained clay and silt. Throughout the early Quaternary, the region underwent subsidence and swift sediment deposition, with coarse sand and gravel prevailing as the chief particulates, indicative of a sedimentary dynamic principally driven by freshwater fluvial processes. Upon analyzing borehole and physical data within the study region, it was noted that the resistivity of geological layers demonstrated a strong correlation with porosity, with strata comprising coarse sand, medium-coarse sand, and sandstone demonstrating relatively high resistivity. However, factors such as water and seawater content were not considered; thus, the lithology’s physical characteristics alone cannot serve as the sole basis for interpreting the profile.

### 5.2. Analysis of Profile Results

Focusing on electromagnetic profile L1 as a case in point, the seismic time-domain offset section of the shallow layer is depicted in Figure 9. In reference to the proximal Xiaoyangkou geothermal well data (outlined in Table 2), ten distinct geological strata can be interpreted from the top to the bottom along the profile. The base interface of the Quaternary (Qh) layer, labelled S1, is inferred to correspond to the uppermost interface of the first confined aquifer, as well as the base interface of the sub-clay layer. S2 represents the Late Pleistocene’s (Qp_3_) base interface, aligning with the bottom of the first confined aquifer and the uppermost layer of the thin clay layer. The mid-clay layer of the second confined aquifer’s base, S3, is interpreted to be a mid-layer of the Lower Miocene (Qp_2_^1^). S4 corresponds to the upper Lower Miocene’s (Qp_1_^3^) sub-clay layer base interface, aligning with the third confined aquifer’s top board interface. S5, the Quaternary’s base interface, also serves as the lower Lower Miocene’s (Qp_1_^1^) base interface. S6, the base interface of the Neogene Yancheng Formation’s (N_1-2_yc) upper and mid sections, is projected to align with the upper Neogene N_2_ segment’s (N_2_^3^) top. S7 designates the upper N_2_ section’s base interface, specifically the N_2_^3^ base interface, and is perceived as the fourth confined aquifer’s mid interface. S8 denotes the mid N_2_ section’s base interface or N_2_^2^’s base interface, which corresponds to the fourth confined aquifer’s bottom plate interface. S9 signifies the lower N_2_ section’s base interface, i.e., N_2_^1^’s base interface, while S10 is identified as the N_1_ base interface, serving as the Neogene’s base interface within the study area, and marking the top interface of the basement. While shallow seismic methodology can accurately depict the geological layers within its resolution limit, its inherent methodological constraints preclude it from delivering accurate interpretations of the distribution of saltwater and freshwater within these geological strata [26,27].

The AMT data presents a stratified arrangement from the superficial to the profound layers, demonstrating a sequence of low resistivity, marginally elevated resistivity, low resistivity, marginally elevated resistivity, and sub-high resistivity. The CSAMT inversion profiles manifest a similar pattern of low resistivity (Figure 10), marginally elevated resistivity, low resistivity, and marginally elevated resistivity, denoting superior resolution in the shallow sections of the CSAMT (Figure 11) [28]. When combined with drilling data, it can be inferred that the stratum spanning from 0 to 20 m correlates with a siltstone layer infiltrated with seawater, characterized by low resistivity (resistivity values ranging from 0.5 to 1.0 Ω·m). The 20–100 m layer corresponds to sandstone and fine sandstone with minor seawater content, exhibiting marginally elevated resistivity (resistivity values falling within 1–10 Ω·m). The 100–200 m layer aligns with coarse sandstone and sandstone presenting high porosity and mineralization, characterized by low resistivity (resistivity values from 0.5 to 1.5 Ω·m). The stratum spanning from 200 to 420 m matches with shale exhibiting low porosity and mineralization, characterized by high resistivity (resistivity values ranging from 2 to 8 Ω·m). Subsequently, layers situated below 420 m correspond to gravel and coarse sandstone layers with minimal water content, thus showing high resistivity (resistivity values ranging from 8 to 100 Ω·m) [29,30,31,32].

### 5.3. Interpretation

Consequently, integrating seismic data with electromagnetic data allows for the formulation of conjectures and elucidations concerning the subterranean architecture and the spatial arrangement of saline and freshwater within the intertidal zone, as depicted in Figure 12.

The principal component of the first stratum exhibits depth variations from 20–50 m, showing modest regional discrepancies and generally trending deeper proximate to the coastline. The demarcation at the base of this layer is postulated to align with the Quaternary’s (Qh) base, mirroring the top boundary of the first confined aquifer (Ⅰ), and marking the base of the sub-clay layer. This stratum primarily functions as the aquifer’s carrier, typified by saline water quality. The inverted apparent resistivity profile manifests this layer as one of low resistivity.

The principal constituent of the second stratum displays a bottom depth predominantly ranging from 80–120 m, observing minimal oscillations across some profiles. The base of this layer is postulated to coincide with the Late Pliocene’s (Qp_3_) bottom boundary, which aligns with the base of the first confined aquifer (Ⅰ) and the superior boundary of the thin clay layer. This stratum is chiefly responsible for carrying the first confined aquifer, typified by brackish water quality. The inverted apparent resistivity profile demonstrates this layer as a relatively high-resistivity stratum.

The primary body of the third stratum exhibits a bottom depth approximating 250 m. The termination of this layer is conjectured to align with the lower boundary of the Upper Pliocene (Qp_1_^3^) sub-clay layer, signifying the superior boundary of the third confined aquifer (Ⅲ). This stratum primarily conveys the second confined aquifer (Ⅱ), marked by a saline water quality and considerable water volume. The inverted apparent resistivity profile reveals this layer as a low-resistivity stratum.

The central structure of the fourth stratum denotes a bottom depth approximating 320 m. The termination of this layer is speculated to coincide with the lower boundary of the Quaternary (Qp_1_^1^), reflecting the third confined aquifer (Ⅲ). This stratum serves as the chief conduit for the third confined aquifer, characterized by fresh water quality. The inverted apparent resistivity profile identifies this as a layer of comparatively high resistivity. The third confined aquifer presents favorable subterranean conditions, with the upper portion composed of brownish-yellow compacted sub-clay, demonstrating stable distribution and substantial thickness. This effectively impedes saline water infiltration from the first and second confined aquifers, thus rendering it an excellent freshwater source within the region.

The core structure of the fifth stratum showcases a pronounced north-deep-south-shallow attribute, with an expansive depth variation fluctuating between 710 m and 910 m. It is postulated that the lower boundary of this layer parallels the terminal boundary of the Neogene (N_1_), concomitant with the upper boundary of the fourth confined aquifer (Ⅳ). This stratum primarily accommodates the fourth confined aquifer, distinguished by its fresh water quality. The inverted apparent resistivity profile delineates this as a layer of high resistivity [33].

## 6. Spatial Distribution of Saline and Fresh Water

Due to the uneven distribution of the survey lines in the plane, Figure 13 only qualitatively analyzes the plane distribution characteristics of each aquifer within a certain range (part of the depth references the shallow seismic results of lines L1, L4, and L5 on the same profile).

Water-bearing layer depth: The primary body of the aquifer in the Xiaoyangkou coastal zone generally exhibits an east-high and west-low trait, with a pronounced depth discrepancy, reaching a maximal depth differential of approximately 27 m.

Depth of the first confined aquifer’s bottom boundary: The depth of this layer in the Xiaoyangkou coastal zone lacks explicit regularity, with a maximum depth difference of approximately 33 m, and the shallowest depth being proximate to the L2 and L4 lines. Predominantly, the lithology of this layer comprises silt, encompassing substantial water volume, but the water quality is subpar, chiefly semi-saline and saline water, surfacing as a relatively high-resistivity layer in the electromagnetic inversion profile.

Depth of the second confined aquifer’s bottom boundary: The depth of this layer in the Xiaoyangkou coastal zone is generally flat, demonstrating an east-high and west-low feature, with a maximum depth difference of approximately 13 m, and a depressed channel near the center-west.

Depth of the Quaternary and third confined aquifers’ bottom boundaries: The Xiaoyangkou coastal zone manifests a west-high and east-low trait, with depths varying from 306 to 332 m. The interpretive results concur with the regional survey report, but with an enhanced resolution and a more discernible depth fluctuation. Additionally, the third confined aquifer presents favorable storage conditions, with a top stratum constituted by brown-yellow dense silt clay, exhibiting stability in its distribution and large thickness. It effectively barricades the saline water from the overlaying first and second confined aquifers, marking it as a premium freshwater layer in the region. Its lithology is primarily fine silt, with some segments incorporating gravel and medium-coarse sand. Consequently, regions with deeper depths of this layer’s bottom boundary in the planar map are advantageous for positioning groundwater pumping wells, given their larger thickness and water content of the aquifer, resulting in an anticipated higher yield from these wells.

## 7. Conclusions

This study focuses on the difficulties of electromagnetic exploration in tidal flat areas and attempts to modify conventional electromagnetic instruments. A water–land amphibious platform dragging high-efficiency data acquisition technology was developed to obtain various electromagnetic data in tidal flat areas. Through two-dimensional inversion processing and combined with other relevant information, the underground structure and distribution of brackish and fresh water in the eastern region of Xiaoyangkou, Nantong, Jiangsu Province were evaluated, providing basic geological data for major engineering construction and freshwater resource development in this area. The main conclusions are as follows:The modified electromagnetic sensors can adapt to the high-salt and highly corrosive environment of tidal flat areas. The water–land amphibious dragging platform can quickly collect data and obtain reliable data. Under the geological conditions of thick and low-resistivity cover in tidal flat areas, the CSAMT detection depth can reach 200 m, and the AMT can reach 500 m. In terms of the resolution of shallow subsurface layers, CSAMT has the stronger detection capability.By combining electromagnetic data with other information, such as shallow seismic exploration, the characteristics of the main structure of the Neogene strata and the distribution of underground brackish and fresh water in the Yangkou Port area can be interpreted. Through the subdivision of the top and bottom boundaries of the main aquifers in the Neogene strata and the thickness of clay aquitards, the salinity and water content of the strata water can be effectively evaluated. The distribution of regional clay aquitards can further analyze the replacement relationship of the quality of fresh water and brackish water in the aquifers.

## Figures and Tables

**Figure 1 sensors-23-06405-f001:**
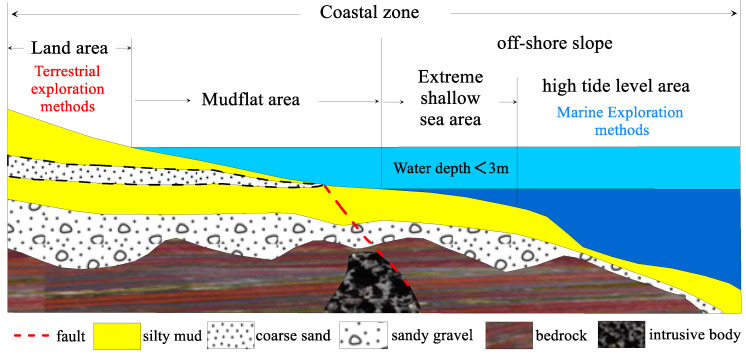
Schematic diagram of the beach area.

**Figure 2 sensors-23-06405-f002:**
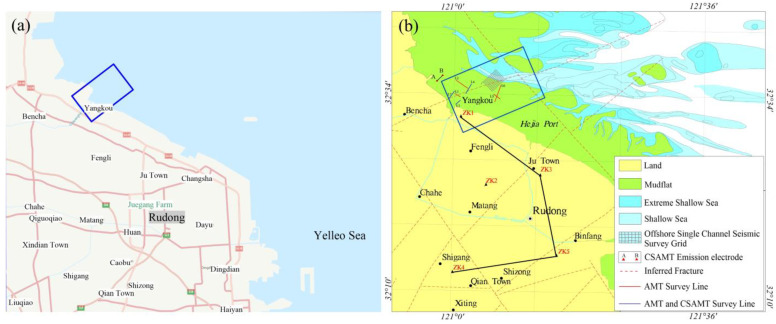
Location of the study area and geophysical geochemical survey ((**a**) location of the study area, (**b**) position of geophysical survey line).

**Figure 3 sensors-23-06405-f003:**
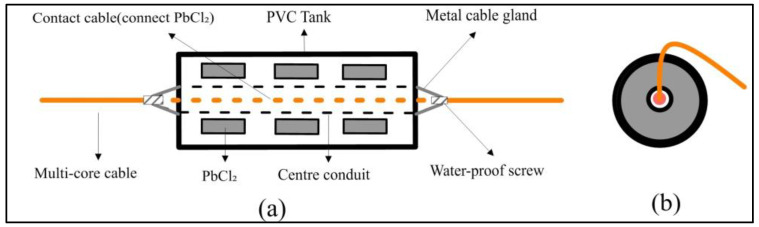
Accumbent multipoint attachment non-polarizing electrode structure ((**a**) cross view, (**b**) top view).

**Figure 4 sensors-23-06405-f004:**
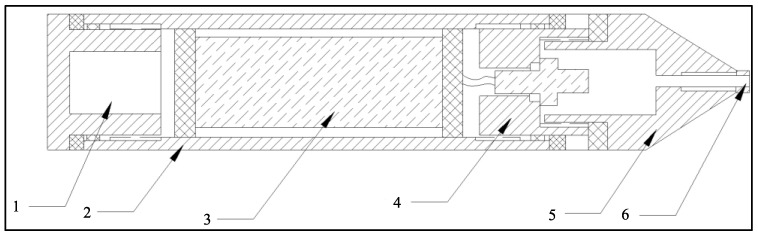
Waterproof magnetic sensor structure (1. bottom waterproof cover, 2. sensor shell, 3. sensor body, 4. top waterproof cover, 5. waterproof cable connector, 6. waterproof screw).

**Figure 5 sensors-23-06405-f005:**
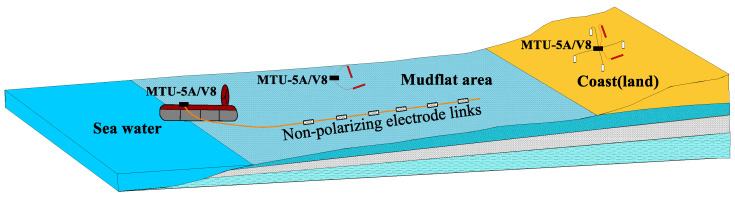
Schematic diagram of the work of the electromagnetic exploration in the beach area.

**Figure 6 sensors-23-06405-f006:**
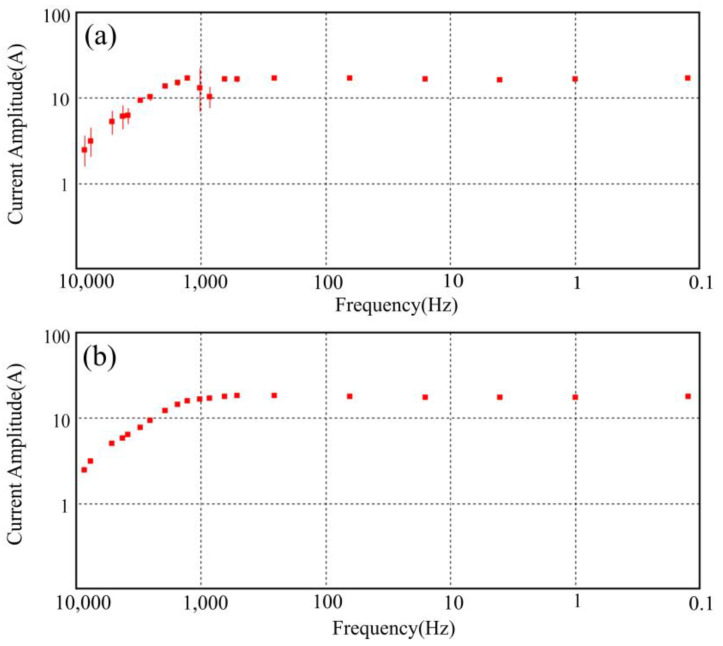
Comparison of CSAMT emission source currents for different grounding conditions ((**a**) grounding conditions not addressed, (**b**) grounding conditions backfill dry soil treatment).

**Figure 7 sensors-23-06405-f007:**
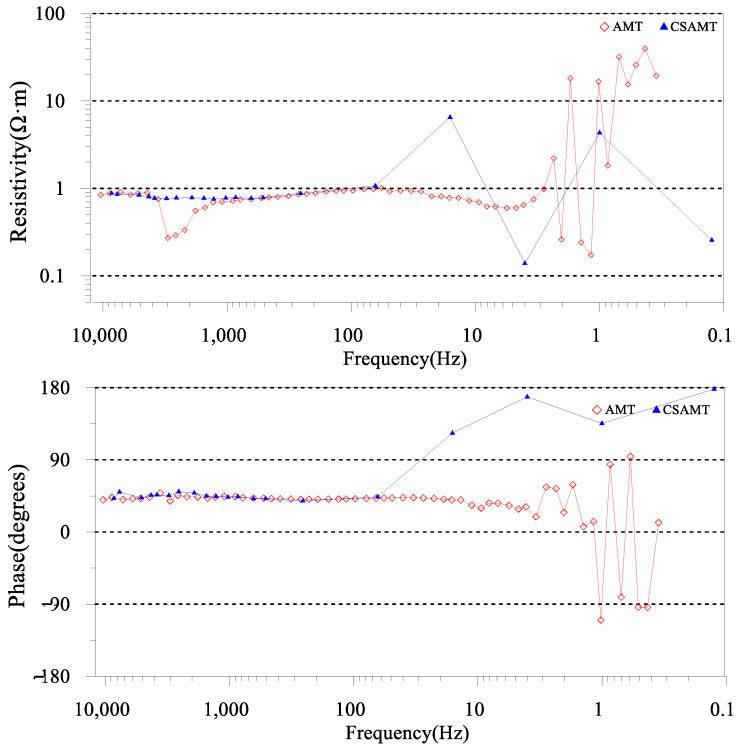
Comparison of apparent resistivity and phase curves of different electromagnetic exploration methods at the beach area measurement points.

**Figure 8 sensors-23-06405-f008:**
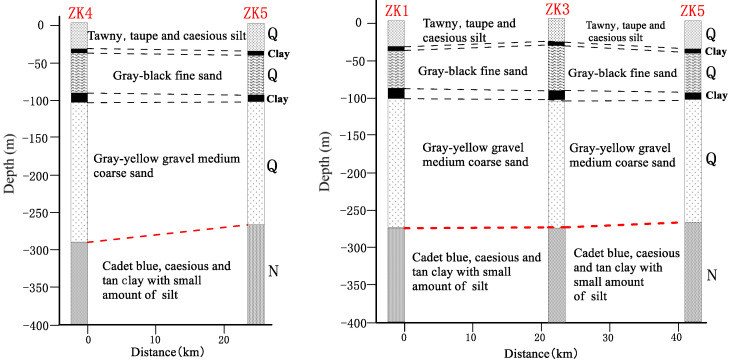
Cross-section of borehole in the study area (according to Jiangsu Changjiang Geological Survey Institute).

**Figure 9 sensors-23-06405-f009:**
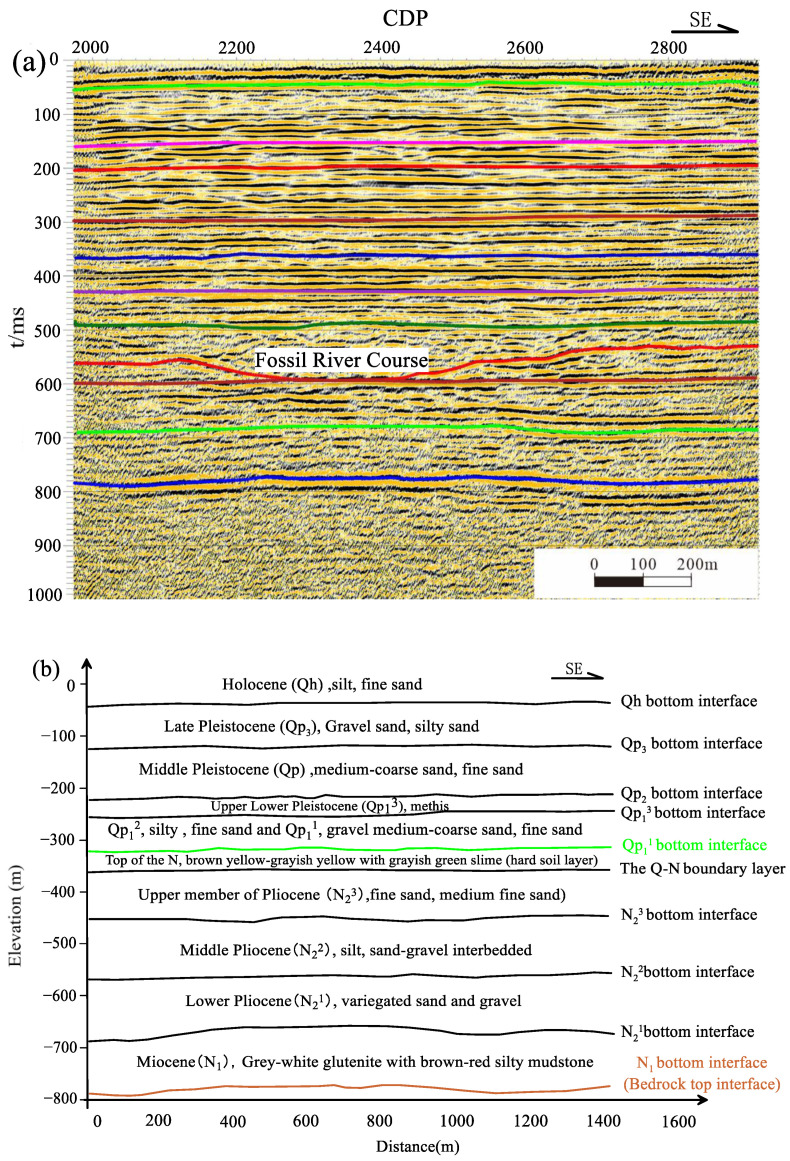
Shallow seismic time profile and stratigraphic layered interpretation profile of L1 line in Xiaoyangkou harbor ((**a**) Shallow seismic time profile, (**b**) layered interpretation profile).

**Figure 10 sensors-23-06405-f010:**
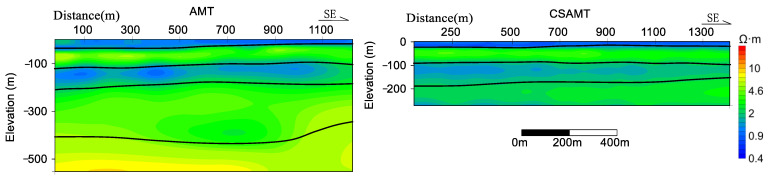
L1 resistivity inversion cross-section of the study area in Xiaoyangkou Port, Rudong.

**Figure 11 sensors-23-06405-f011:**
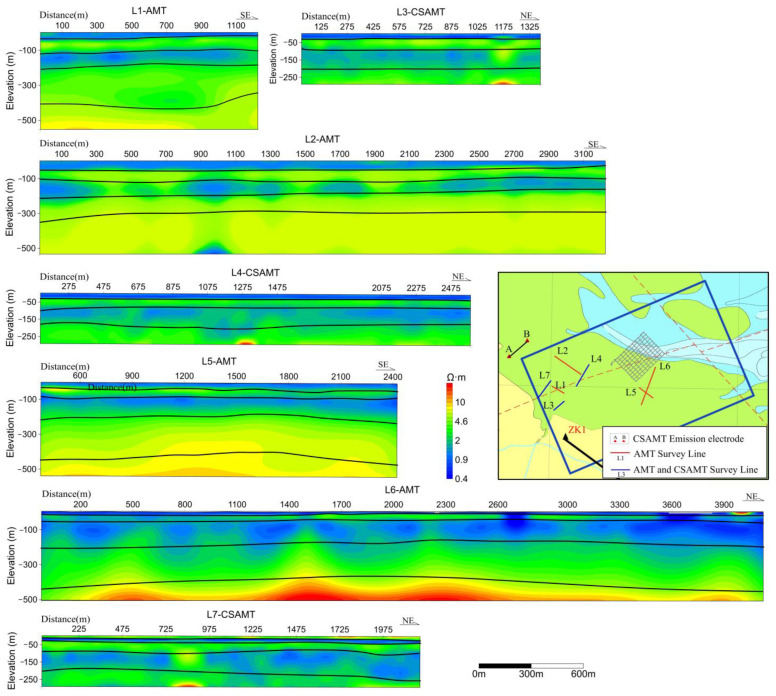
Extrapolation of the general inversion interpretation of the electromagnetic exploration in the Xiaoyangkou port area.

**Figure 12 sensors-23-06405-f012:**
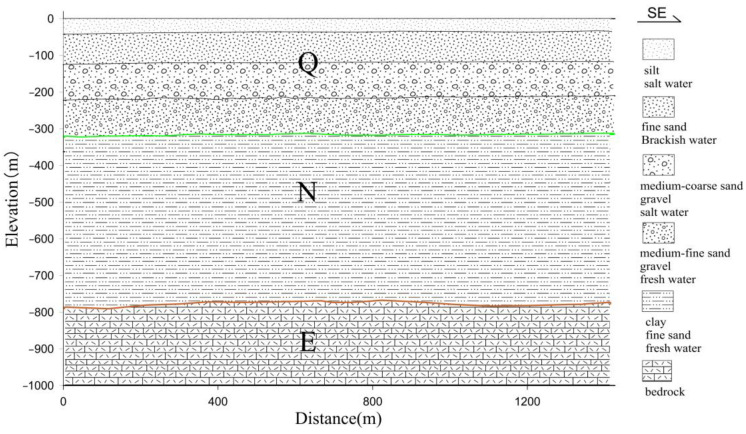
Integrated interpretation inferred map of line L1.

**Figure 13 sensors-23-06405-f013:**
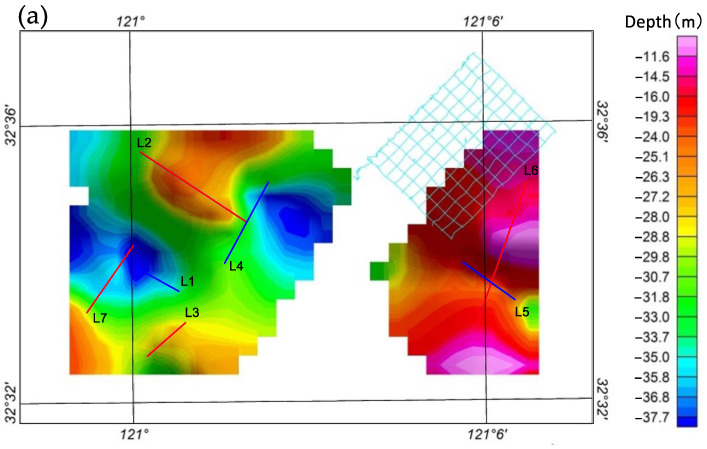
Planar distribution characteristics of each water-bearing layer: (**a**) Burial depth plan of the unconfined aquifer layer bottom boundary; (**b**) Burial depth plan of the Ⅰ pressurized aquifer bottom boundary; (**c**) Burial depth plan of the II pressurized aquifer bottom boundary; (**d**) Burial depth plan of the quaternary bottom boundary.

**Table 1 sensors-23-06405-t001:** Correspondence between regional electrodeposition ρ_s_ values and stratigraphy.

Areas	Electrical Layer	ρ_s_/(Ω·m)	Geological Properties
Freshwater areas	ρ_1_	10–30	Quaternary topsoil layer
	ρ_2_	20–60	Quaternary sand and clay interbeds
	ρ_3_	10–20	Quaternary bottom clay, Neoproterozoic mudstone
	ρ_4_	5–15	Paleocene to Middle Triassic
	ρ_5_	>200	Paleozoic
Saline water areas	ρ_1_	10–30	Quaternary topsoil layer
	ρ_2_	2–12	Quaternary, Neoproterozoic, and Paleocene sands
	ρ_3_	15–50	Cretaceous to Middle Triassic
	ρ_4_	>200	Paleozoic

**Table 2 sensors-23-06405-t002:** Rock chip logging table of geothermal well No. 1 in Xiaoyangkou [19].

Stratigraphic	Starting Depth (m)	Final Depth (m)	Thickness (m)	Lithological Description
Series 4 Q	0	15	15	Silt
	15	30	15	Coarse-grained sand
	35	100	65	Light grey fine sand, quartz sand
	100	125	25	Large-grained sandstone, ginger gravel, pale white quartz sand
	125	165	40	Large grains of sea sand
	165	250	85	Miscellaneous sandstone
	250	290	40	Grey mudstone, medium-grained sandstone
	290	301	11	Coarse-grained sea sand
	301	304	3	Coarse sand
	306	316	10	Light yellow clay with minor sandstone
	318	324	6	Grey soft mudstone, sandstone
Neophyte system N	324	336	12	Light yellow clayey mud, light grey sandstone, conglomerate
	338	384	46	Light grey muddy sandstone, fine sandstone
	386	398	12	Yellow soft mudstone, coarse-grained sandstone
	400	406	6	Grey muddy sandstone
	408	430	22	Grey clayey mud, large-grained sandstone, gravel
	432	444	12	Light yellow mudstone, coarse-grained sandstone, gravel
	446	464	18	Greyish yellow mudstone, quartz sand, gravel
	466	472	6	Large-grained sandstone, quartz sand, gravel
	474	498	24	Greyish light grey coarse-grained, fine-grained sandstone, gravel, quartz sand
	502	510	8	Grey clay, coarse sand, fine sand, gravel
	512	526	14	Light yellow clayey mud, coarse-grained sandstone, gravelly with more gravel
	528	538	10	Large gravel
	540	564	24	Light yellow soft mudstone, large-grained quartz sand
	566	598	32	Light grey clay, fine sand
	598	610	12	Grey clay, fine sand
	612	616	4	Coarse-grained quartz sand
	618	626	8	Yellow soft clay, large-grained quartz sand
	628	646	18	Grey clay, medium to coarse quartz sand
	648	686	38	Large to medium coarse-grained quartz sand with nodule agglomerates
	688	688.5	0.5	Light yellow clay, large quartz sand
	690	698	8	Soft yellow clay, a little grit
	700	732	32	Light grey soft clay, muddy sand
	734	746	12	Dark red soft clay, large-grained quartz sand
	748	758	10	Grey clay sand, coarse-grained quartz sand
	760	760.5	0.5	Grey soft clay sand
	762	762.5	0.5	Brownish-black hard mud slate, coarse-grained quartz sand
	764	776	12	Brownish-black carbonaceous rigid mudstone, pinkish-white calcareous siltstone, coarse-grained quartz sand, sandstone with a little gravel
	788	794	6	Light yellow-grey muddy sand, gravel
Paleocene E	796	818	22	Black carbonaceous hard mudstone, minor off-white calcareous sandstone, large-grained quartz sand
	828	840	12	Light yellow soft clay, silt
	842	848	6	Light grey fine sandstone, clay, quartz sand
	850	856	6	Light red soft mudstone, greyish white calcareous sandstone, quartz sand, gravel
	858	866	8	Quartz sand, pale yellow gravel, minor sand grains, transformed to brownish-black hard mud slate, quartz sand, minor sand grains
	868	1008	140	Yellow sandstone, gravel, quartz sand, fine sand, brown hard mudstone
	1010	1018	8	Pink mudstone, quartz sand, fine sand, gravel, brown hard mudstone
	1020	1058	38	Gravel
Cretaceous K	1058	1073	15	Sandstone

Note: Final hole depth 1073 m.

## Data Availability

Not applicable.

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
