# Peer review of "Application of the Electromagnetic Method to the Spatial Distribution of Subsurface Saline and Fresh Water in the Coastal Mudflat Area of Jiangsu Province"

_sensors, 2023, doi:10.3390/s23146405_

Round 1

Reviewer 1 Report

The author focuses on the challenges of electromagnetic exploration in tidal flat areas and presents a modified approach using a water-land amphibious platform with high-efficiency data acquisition technology. The study evaluates the underground structure and distribution of brackish and fresh water in the eastern region of Xiaoyangkou, Nantong, Jiangsu Province. The modified electromagnetic sensors and dragging platform are found to adapt well to the high-salt and corrosive environment of tidal flat areas, providing reliable data. The study demonstrates that the CSAMT detection depth can reach 200m and the AMT can reach 500m in tidal flat areas with thick and low-resistivity cover. The CSAMT method also exhibits stronger detection capability for shallow subsurface layers. The findings contribute to the understanding of the geological conditions in the study area and provide valuable data for major engineering construction and freshwater resource development. Detail comments see attached file.

The author demonstrates a commendable command of the English language, with the manuscript largely well-written and articulate. Nonetheless, minor editing is requisite to refine the language further. Notably, there are occasional instances where sentence structure could be optimized for clarity, and certain terminologies may be refined to align more closely with standard scientific parlance. Additionally, attention to grammatical nuances and syntactical flow would be beneficial in ensuring that the manuscript meets the high standards expected in scholarly communication. Overall, these minor revisions will not alter the content significantly but will enhance the readability and professional polish of the manuscript.

Reviewer 2 Report

The unique terrain and geological differences in the muddy area sandwiched between land and ocean pose significant challenges to the application of electromagnetic methods. The manuscript attempts to innovate and improve existing instruments, and adopts electromagnetic data acquisition technology suitable for muddy environments. Based on the CASAMT and AMT data, combined with the interpretation of multiple profiles, the electrical characteristics of the muddy aquifer were explored, and the spatial distribution information of the underground structure and fresh and saline water resources in the detection area was provided. The manuscript has obvious novelty and practical value, and I am very happy to be able to read it. Suggest publishing after moderate modifications.

1. It seems that there is no textual description of Figures 1 and 2 in the text. It is best to annotate or add Caption explanations for the layers corresponding to different colors in Figure 1

2. In section 2.3, regarding ρ 1 and ρ 2. The description of equivalent variables should correspond to Table 1, otherwise it may cause misunderstandings;

3. Is there a traction cable connecting multiple electrodes during the potential data collection process for the developed non-polarized electrode receiving array? If so, how many electrodes are there in an arrangement? What is the distance between the arranged electrodes?

4. Both CSAMT and AMT observations were conducted, and it is recommended to provide more observation related parameters. For example, the transmitting-receiving distance? What is the emission current? What adjustments have been made to the low resistivity observation environment of the mudflat?

5. Figure 6, (H) should be f (H). The equations (3) and (4) in the text should annotate the meaning of each variable.

6. The color in Figure 12 (c) is marked incorrectly and needs to be modified.

7. The format of the reference citation needs to be modified.

 Minor editing of English language required
